**Data Availability Statement:** All relevant data are within the paper and its Supporting information files.

# Human T-Lymphotropic virus type 1 and human immunodeficiency virus co-infection in rural Gabon

**Augustin Mouinga-Ondémé**[1]*, **Larson Boundenga**[2,3], **Ingrid Précilya Koumba Koumba**[1,4], **Antony Idam Mamimandjiami**[1], **Abdoulaye Diané**[1], **Jéordy Dimitri Engone-Ondo**[1], **Delia Doreen Djuicy**[1], **Jeanne Sica**[5], **Landry Erik Mombo**[4], **Antoine Gessain**[6], **Avelin Aghokeng Fobang**[7,8]

1 Unité des Infections Rétrovirales et Pathologies Associées, Centre International de Recherches Médicales de Franceville (CIRMF), Franceville, Gabon, 2 Groupe Evolution et Transmission Inter-espèces des Pathogènes (GETIP), Département de Parasitologie, Centre International de Recherches Médicales de Franceville (CIRMF), Franceville, Gabon, 3 Department of Anthropology, Durham University, Durham, United Kingdom, 4 Laboratoire de Biologie Moléculaire et Cellulaire (LABMC), Université des Sciences et Techniques de Masuku (USTM), Franceville, Gabon, 5 Centre de Traitement Ambulatoire, Franceville, Gabon, 6 Unité d'Epidémiologie et de Physiopathologie des Virus Oncogènes (EPVO), et CNRS UMR3569, Institut Pasteur de Paris, Paris, France, 7 Unité Mixte de Recherche sur le VIH et les Maladies Infectieuses Associées, Centre International de Recherches Médicales de Franceville (CIRMF), Franceville, Gabon, 8 MIVEGEC, Université de Montpellier, CNRS, IRD – Montpellier, Montpellier, France

* ondeme@yahoo.fr

## Abstract

### Introduction

Human T-cell lymphotrophic virus type-1 (HTLV-1) and human immunodeficiency virus (HIV-1) co-infection occur in many populations. People living with HIV-1 and infected with HTLV-1 seem more likely to progress rapidly towards AIDS. Both HTLV-1 and HIV-1 are endemic in Gabon (Central Africa). We investigated HTLV-1 and HIV-1 co-infection in the Haut-Ogooué province, and assessed factors that may favor the rapid evolution and progression to AIDS in co-infected patients.

### Methods

Plasma samples from HTLV-1 patients were tested using ELISA, and positive samples were then tested by western blot assay (WB). We used the polymerase chain reaction to detect HTLV-1 Tax/Rex genes using DNA extracted from the buffy coat of ELISA-positives samples.

### Results

We recruited 299 individuals (mean age 46 years) including 90 (30%) men and 209 (70%) women, all of whom are under treatment at the Ambulatory Treatment Centre of the province. Of these, 45 were ELISA HTLV-1/2 seropositive. According to WB criteria, 21 of 45 were confirmed positive: 20 were HTLV-1 (44%), 1 was HTLV-1/2 (2%), 2 were indeterminate (4%) and 22 were seronegative (49%). PCR results showed that 23 individuals were positive for the Tax/Rex region. Considering both serological and molecular assays, the

**Funding:** The authors received no specific funds for this work.

**Competing interests:** The authors have declared that no competing interests exist.

prevalence of HTLV-1 infection was estimated at 7.7%. Being a woman and increasing age were found to be independent risk factors for co-infection. Mean CD4+ cell counts were higher in HTLV-1/HIV-1 co-infected (578.1 (± 340.8) cells/mm$^3$) than in HIV-1 mono-infected (481.0 (± 299.0) cells/mm$^3$) Individuals. Similarly, the mean HIV-1 viral load was Log 3.0 (± 1.6) copies/ml in mono-infected and Log 2.3 (± 0.7) copies/ml in coinfected individuals.

## Conclusion

We described an overall high prevalence of HTLV-1/HIV-1 co-infection in Gabon. Our findings stress the need of strategies to prevent and manage these co-infections.

## Introduction

Human T-Lymphotropic virus type 1 (HTLV-1) was the first HTLV to be discovered and is one of the most prevalent human retroviruses in the world [1, 2]. This oncoretrovirus is highly endemic in some restricted regions of Southwestern Japan, Iran, Australo-Melanesia, sub-Saharan Africa, Caribbean basin and South American [1]. It is estimated that at least 5 to 10 million persons are infected around the world and that 2% to 7% of them develop related pathologies [3, 4]. HTLV-1 is the etiological agent of several pathologies, such as a very severe T-cell lymphoproliferation named Adult T-Cell Leukemia Lymphoma (ATLL) and a progressive disabling neuro-myelopathy, the Tropical Spastic Paraparesis/HTLV-1 Associated Myelopathy (TSP/HAM) [2, 5–7]. HTLV-1 can be transmitted through unprotected sexual intercourse, particularly from men to women [8], prolonged breastfeeding from an infected mother to her child [9], and contaminated blood products, during transfusion or intravenous drug injections [10, 11]. HTLV-1 shares transmission routes with Human Immunodeficiency virus (HIV); moreover, the two retroviruses have a common *in vivo* tropism, since CD4+ T-cells are their major targets of infection [12].

At the end 2019, the World Health Organization (WHO) estimated that 38 million people were living with HIV infection in the world [13]. Without treatment, HIV-infected individuals develop Acquired Immunodeficiency Syndrome (AIDS) and may die [14]. Previous studies have reported HTLV-1 and HIV-1 co-infections [15–18] with a prevalence estimated at 0.11% to 10.9% depending on the geographic region and group studied [16, 19–21]. However, there are suggestions that people living with HIV-1 and infected with HTLV-1 are more likely to progress rapidly towards AIDS [22–24]. Moreover, the prevalence of HAM/TSP increases in HIV-1/HTLV-1 coinfected individuals [15, 25].

HTLV-1 and HIV-1 are very prevalent in Gabon. HTLV-1 seroprevalence ranges from 5% to 10.5% in the adult rural population [26, 27] and is 0.7% in blood donors [11]. Similarly, during the last Demographic and Health Survey (DHS), HIV-1 infection was been reported in 4.1% of general population aged 15 years to 49 years [28]. The distribution of HTLV-1 and HIV-1 across the country showed that the Haut-Ogooué province, in the south-east part of the country, is one of the most affected, with 9.5% to 11.6% of adult people living with HTLV-1 [26, 27] and 4.2% with HIV-1 [28]. Despite such very high seroprevalence for both retroviruses, no studies have been published, to our knowledge, on co-infections in populations living in Gabon.

To extend our knowledge of this subject, we investigated HTLV-1 infection in a population of HIV infected patients living in the southern region of Gabon and assessed factors that may favor the rapid evolution and progression to AIDS in such co-infected patients.

## Methods

### 1. Study design and population

We conducted a cross-sectional study from June 2018 to September 2019 in the southern region of the country to investigate HTLV-1 and HIV-1 co-infection in a cohort of patients followed-up and monitored at the major Ambulatory Treatment Centre (CTA) of Franceville (a city of 7,000 inhabitants). The recruitment criteria were as followed: (i) being confirmed HIV-1-positive; (ii) being on antiretroviral therapy (ART), (iii) and provide informed consent. We collected the following information: gender, age, ART regimen, last CD4 counts at ART initiation, and HIV-1 viral load (VL) results obtained as previously described [29].

The study was approved by the National Ethics Committee, registered as PROT N˚0011/ 2013/SG/CNE and all participants gave verbal consent to participate in the study.

### 2. HIV viral load and CD4 counts

Blood specimens were collected from each participant using 5 ml EDTA tubes and transferred to a reference national laboratory, the CIRMF Retroviral Infection Laboratory. Samples were processed on arrival at the laboratory and plasma and buffy coat layers were collected and stored at -80˚C until testing. CD4 counts at enrolment were determined using flow cytometry (Fascount, Becton Dickinson, San Jose, CA). Plasma HIV-1 VL was determined using the Generic HIV Viral Load test (Biocentric, Bandol, France) performed according to the manufacturer's instructions, using the protocol with a detection limit of 300 copies/mL and 200 ml of plasma.

### 3. HIV diagnosis

HIV-1 RNA was extracted from the plasma using the QIAamp RNA Viral Mini kit (Qiagen, Courtaboeuf, France). Reverse transcriptase HIV fragment was amplified by RT-PCR using SuperScript III One-Step RT–PCR (Thermo Fisher Scientific, Waltham, MA, USA), and the primers pair RT18/RT21, followed by a second-round PCR using Invitrogen Platinum Taq DNA Polymerase (Thermo Fisher Scientific) with RT1/RT4 primers. Genotypic HIV-1 drug resistance testing was performed using the ANRS (French National Agency for Research on AIDS and Viral Hepatitis) version 2015 protocol, targeting the protease (PR) and reverse transcriptase (RT) regions (http://www.hivfrenchresistance.org).

### 4. HTLV diagnosis

To determine HTLV infection, we tested plasma samples using an ELISA assay (HTLV-1/2 ELISA 4.0 MP Biomedicals, Singapore) for the initial screening and detection of HTLV-1/2 antibodies.

The reactions and interpretation of the results were performed according to the manufacturer's instructions. The cut-off value was calculated as (0.25 + non-reactive control mean absorbance).

All ELISA positive samples were tested with a Western Blot (WB) assay (HTLV Blot 2.4 MP Biomedicals, Singapore) to confirm and discriminate between HTLV-1 and HTLV-2 infections.

Briefly, HTLV-1-positive plasma samples were defined as the presence of gag (p19 with/ without p24) and two env (GD21 and rgp46-I) bands. Likewise, HTLV-2-positive samples were defined as reactive to gag (p24 with/without p19) and two env (GD21 and rgp46-II) bands. Samples displaying antibodies to both gag (p19 and p24) and env (GD21) bands were defined as HTLV-positive but untypeable. Any other patterns of bands were denoted as

indeterminate. For all samples with a positive ELISA result, we extracted viral DNA from buffy-coat specimens using the Qiagen DNA Blood Mini kit (Qiagen, Courtaboeuf, France). DNA concentration was evaluated with a Nanodrop spectrophotometer (Thermo Fisher Scientific®), and its integrity was verified by amplifying an Albumin gene fragment, as described previously [30]. HTLV DNA amplification was conducted using a generic PCR approach with consensus primers amplifying the PTLVs *tax* region of 220 bp, as previously described [31]. Briefly, for the first round, the *tax* primers used were AV46 and AV45 to amplify the region of 303 bp based on 250 ng of genomic DNA. Then, a 220 bp fragment was amplified during the second round with nested primers AV42 and AV43. All *tax*-positive samples were further analyzed with another semi-nested PCR to amplify a 522 bp fragment of the Env gene with the Env11 and Env22 primers, as previously described [32]. Amplicons of the expected size were sent to Macrogen Europe B.V (Meibergdreef 57, 1105 BA Amsterdam, Pays-Bas) for sequencing.

### 5. Phylogenetic analysis

HTLV-1 genotypes were identified by BLAST searches of the sequences obtained, using the GenBank database (http://www.ncbi.nlm.nih.gov/BLAST). A phylogenetic tree was generated from the alignment of the 522 bp Env gene fragment sequences obtained, by Neighbor-Joining method, with MEGA 7 software.

### 6. Statistical analysis

Statistical analyses were performed using R software. HTLV-1 infection prevalence was compared with gender using a Chi-square ($\chi2$) test. We compared HTLV-1 infection prevalence and age groups using logistic regression. We compared the mean of CD4+ lymphocytes count between mono-infected (HIV-1 only) and co-infected (HIV-1/HTLV-1) individuals using a Chi-square ($\chi2$) test. Finally, we compared the mean of viral load between mono-infected and co-infected individuals using a nonparametric Wilcoxon test. The statistical significance was defined when a *p*-value was $<0.05$.

## Results

### Patient population

Overall, we recruited 299 individuals including 209 (70%) women and 90 (30%) men (Table 1). The median age was 46 years with an interquartile range (IQR) between 34 and 58 years. The median CD4-cell count was 468 (IQR, 256–692) at the time of the inclusion in the study. Viral load was available for all patients and the median VL was Log 2.1 (IQR, 1.7–3.8). All participants were receiving first-line antiretroviral treatment (ART) and the mean time on ART was 17 months. Predominant ART regimens included zidovudine (AZT) plus lamivudine (3TC) plus nevirapine (NVP)/efavirenz (EFV) (48%), and tenofovir (TDF) plus 3TC/emtricitabine (FTC) plus EFV (46%) (Table 1).

### Prevalence of HTLV-1 infection

Of the 299 plasma samples tested with ELISA, 15% (45/299) were HTLV-1/2 seropositive. Western Blot analyses confirmed HTLV infection for 21 (7%) specimens; HTLV-1 infection was confirmed for 20 participants and one individual was HTLV-1/2 co-infected. Two (4%) samples displayed indeterminate profiles and 22 (49%) specimens which scored positive on ELISA were considered HTLV negative according to the WB results. Molecular investigations were performed on DNA extracted from buffy coat of all 45 ELISA-positive individuals.

**Table 1. Patient characteristics associated with HTLV-1/HIV-1 coinfection status according to serology and molecular assays.**

| Variables | N = 299 | HTLV-1/2 Serology | | HTLV-1 PCR (Tax/Rex) |
|---|---|---|---|---|
| | | ELISA n [% (95% CI)] | WB n [% (95% CI)] | Positive n [% (95% CI)] |
| **Gender**, n (%) | | | | |
| Men | 90 (30.1) | 13 [14.4(10–19.8)] | 3 [3.3 (-0.1–6.7)] | 3 [3.3 (-0.1–6.7)] |
| Women | 209 (69.9) | 32 [15.3 (10.3–20.3)] | 18 [8.6 (4.8–12.4)] | 20 [9.6 (5.6–13.6)] |
| Total | 299 (100) | 45 [15.1 (10.2–21)] | 21 [7.02 (3.3–11.6)] | 23 [7.7 (3.9–12.1)] |
| **Age**, n (%) | | | | |
| 18–30 y | 52 (17.4) | 7 [13.5 (9–23)] | 3 [5.8 (-1-12.6)] | 3 [5.8 (-1-12.6)] |
| 31–49 y | 129 (43.1) | 16 (12.40±5.9) | 7 [5.4 (1.7–9.1)] | 8 [6.2 (2.1–10.3)] |
| 50–80 y | 118 (39.5) | 22 [18.8 (11.7–25.9)] | 11[9.3 (4–14.6)] | 12 [10.2 (4.7–15.7)] |
| **Median (IQR) 46 (34–58)** | | | | |
| **CD4 cell count, cells/mm$^3$ n (%)** | | | | |
| <200 | 56 (18.7) | 6 [10.7 (1.9–19.5)] | 3 [5.4 (-0.6–11.4)] | 3 [5.4 (-0.6–11.4)] |
| 200–499 | 108 (36.1) | 18 [16.7 (9.6–23.8)] | 6 [5.6 (1.2–10)] | 6 [5.6 (1.2–10)] |
| ≥500 | 135 (45.2) | 21 [15.6 (9.6–21.6)] | 12 [8.9 (5.2–13.6)] | 14 [10.4 (5.3–15.5)] |
| **Median (IQR) 468 (256–692)** | | | | |
| **Virological outcome, n (%)** | | | | |
| Viral load undetectable | 195(65.2) | 32 [16.4 (11.3–21.5)] | 18 [09.2 (5.2–13.2)] | 19 [09.7(5.6–13.8)] |
| Viral load ≥300 copies/ml* | 7(2.3) | 1 [14.3 (5.6–23)] | 1 [14.3 (5.6–23)] | 1 [14.3 (5.6–23)] |
| Viral load ≥1000 copies/ml | 97(32.5) | 12 [12.4 (5.2–19.6)] | 2 [2.1 (-0.9–5.1)] | 3 [3.1 (-0.2–6)] |
| **Median** ** **(IQR) 2.1 (1.7–3.8)** | | | | |
| **ART regimen** | 273/299 | | | |
| AZT + 3TC + EFV, n (%) | 107 (39.2) | 20 ([18.7 (11.3–26.1)] | 6 [5.6 (1.2–10)] | 6 [5.6 (1.2–10)] |
| TDF + 3TC + EFV | 46 (16.8) | 7 [15.2 (4.725.75)] | 4 [8.7 (0.5–16.9)] | 4 [8.7 (0.5–16.9)] |
| AZT + 3TC + NVP | 24 (8.8) | 4 [16.7 (1.5–31.9)] | 2 [8.3 (1.1–15.5)] | 2 [8.3 (1.1–15.5)] |
| TDF + FTC + EFV | 79 (28.9) | 10 [12.7 (5.3–20.1)] | 4 [5.1 (0.2–10)] | 5 [6.3 (0.9–11.7)] |
| Others | 17 (6.2) | 6 [35.3 (11.9–58.7)] | 5 [29.4 (9.1–49.7)] | 6 [35.3 (11.9–58.7)] |

*300 copies/ml is the assay lower detection limit.

**The median viral load level was calculated for results ≥300 copies/ml

3TC: Lamivudine, EFV: Efavirenz, TDF: Tenofovir,

AZT: Zidovudine, FTC: Emtricitabine, NVP: Nevirapines

Generic PCR results showed that a PTLVs *tax* region was successfully amplified for 23 specimens (8%) the 20 HTLV-1 positive individuals in WB, the HTLV-1/2 co-infected individual and the two indeterminate individuals (WB reactivity was shown only for GD21 and P19). Thus, the combination of serological and molecular testing led to an estimated overall HTLV-1 prevalence of 7.7%. All data are summarized in Table 1.

## Phylogenetic analysis

Blast comparative and phylogenetic analyses of the 522bp fragment in the *env* gene showed that 13 of the new 14 characterized strains belonged to the Central African HTLV-1b genotype. Only one strain belonged to the HTLV-1d genotype. Within the HTLV-1b subclade, new strains grouped into two groups (S1 Fig).

### Risk factors for HTLV-1 infection in the study population

Women were more likely to be HTLV-1 infected than men: 20 of 209 women (9.6%) and 3 of 90 men (3.3%) were HTLV-1 infected, but this difference was not statistically significant ($p > 0.05$). Seroprevalence gradually increases with age in this study population ($p < 0.01$). Adult individuals of the 50–80 y (10%) and the 31–49 y (6.2%) age groups are at increased risk of being HTLV-1 seropositive compared to younger people of the 18–31 y age group (5.8%) (Table 1, S2A Fig).

Clinical parameters showed higher levels of CD4+ T cells in HTLV-1/HIV-1 co-infected individuals than in HIV-1 mono-infected: respectively 578.1 (273.3–918.3) cells/mm$^3$ and 481.0 (182–780) cells/mm$^3$. However, this difference was not statistically significant ($p = 0.28$). The mean viral loads were Log 3.0 (Log1.4-Log4.6) copies/ml and Log 2.3 (Log1.6-Log3.0) copies/ml in mono-infected and co-infected individuals, respectively, a statistically significant difference ($p = 0.02$) (Table 1).

Thirteen of 14 HIV-1/HTLV-1 co-infected patients had a viral load suppression (VL<2.5 Log); Only one patient had a virological failure (Gab-2276FCV, co-infected with the HTLV-1D genotype). The HIV-1 RT region of 646 bp infecting this patient was successfully sequenced. This patient was receiving ABC+3TC+EFV first-line antiretroviral therapy. Several HIV drug resistance mutations (DRMs) were identified: L74I, V75T, Y115F, M184V (NRTI), K103N, Y181C (NNRTI).

## Discussion

Previous studies of HTLV-1 conducted in Gabon focused on infection in different types of population including pregnant women, blood donors, pygmy people and rural Bantou communities [11, 26, 27, 33]. We report here HTLV-1/2 infection among people living with HIV/AIDS (PLWHA) in the southeast of the country. Most of the study participants were women. This was expected as most HIV-1 infected patients receiving ART in the Ambulatory Treatment Centre (CTA) in Gabon are women [29, 34]. We found that HTLV-1 is highly endemic among PLWHA. Women were at higher risk of HTLV/HIV co-infection compared to men. The infection increases with age but no significant difference was found in the CD4+ T-cell count between HTLV-1/HIV-1 co-infected individuals and those mono-infected with HIV-1. The mean viral load was significantly higher in HIV-1 mono-infected than in HTLV-1/HIV-1 co-infected individuals, although one should interpret this with caution, since patients were all on ART. Most HTLV-1 strains from HIV-1/HTLV-1 co-infected patients belonged to the central African HTLV-1b genotype. This genotype is the most frequent in Gabon [11]. The unique HTLV-1d case found confirms that this rare genotype is occasionally reported in this country [27].

The prevalence of 7.7% for HTLV-1/HIV-1 co-infection, found in this study, confirms the high rate of HTLV-1 infection in Gabon and particularly in the rural region of Franceville and around. This prevalence is in the range of that reported in rural adults at the national level in Gabon, estimated from 7.3% to 8.7%, including recent data reported in the Haut-Ogooué province ranging from 9.5% to 11.6% [26, 27].

This high rate of HTLV-1/HIV-1 co-infection is an important public health concern in a country like Gabon where both infections are endemic. Infection with HTLV-1 can stimulate an abnormal increase of CD4+ T-cells which may bias ART strategies as CD4+ is a major immunological marker in HIV/AIDS monitoring and lead to a delay in ART initiation in regions where the WHO "test and treat" approach for HIV-1 infection is still to be implemented [15, 20]. Moreover, the consequence of the virological failure is the emergence of mutations that could induce ART drug resistance to the virus. The description of the M184V

mutation in a HTLV-1/HIV-1 co-infected patient (Gab-2276FCV) treated with 3TC confirms the replication of HIV-1 and potentially the HTLV-1 one [35, 36]. HTLV-1/HIV-1 co-infected individuals are predisposed to develop neurological diseases, which are hardly investigated in resource-limited settings, but which may represent a significant public health concern [16]. Moreover, a study of 29 coinfected patients from Mozambique (representing 4.5%) of a series of 704 HIV/AIDS patients showed that co-infected patients exhibited higher levels of CD4+ T cells expressing activation markers and a massive loss of naïve cells, suggesting that co-infected patients may progress faster to AIDS [24].

Several factors are involved in HTLV-1 transmission and probably to the evolution to related diseases in mono-infected and HIV-1/HTLV-1 co-infected individuals. Clinical aspects and outcomes in persons co-infected with both retroviruses were recently summarized in the review [37]. And as previously shown, co-infection with both retroviruses may lead to rapid evolution to AIDS and development of opportunistic infections in the absence of ART [38, 39]. Similarly, higher rates of the Tropical Spastic Paraparesis/HTLV-1 Associated Myelopathy (HAM/TSP) and peripheral neuropathy were observed in contexts of HIV-1/HTLV-1 co-infections [16], and higher mortality rates were also reported [40–42].

We observed, in our study, a significant increase of HTLV-1 infection with age and with a high prevalence of HTLV-1 in women. Similar findings have been reported for a rural area in Gabon [26, 27]; furthermore, most of the co-infected patients in our study were aged from 50 to 80 years. More importantly, in the same place, at Franceville, HTLV-1 prevalence among HIV-1 infected women was higher (9.6%) than among those from the general population (4.4%). In contrast, this prevalence was lower among HIV-1 infected men (3.3%) comparatively to men in general population (4.7%) (S2B Fig). Data from the general population was extracted from the paper by Djuicy et al [26]. A similar result was found in Brazil [21]. This observation of the increased rate of infection with age especially in women is considered as a specificity of HTLV-1 virus [26]. These findings have been attributed to sexual transmission of HTLV-1, especially from males to females [43, 44]. The high rate of HIV-1/HTLV-1 co-infection in women also highlights the risks of vertical transmission of these viruses. Efficient programs have been developed over the past 4 decades to prevent the vertical transmission of HIV from mother to child, especially by using antiretroviral drugs [45]. However, although mother to child transmission is currently recognized as a major transmission route for HTLV-1, there is still no program developed to address this issue in most African countries. Our results indicate that such programs should be urgently developed and implemented. Also, none of the study participants knew they were infected with HTLV before we conducted this study. In a country like Gabon, where both viruses are known to circulate at high prevalence, routine testing for both infections should be considered to help develop prevention and care strategies.

Although the prevalence of these two retroviruses is very high in many African countries, limited number of studies have been carried on the co-infection. Studies of different types of population in African countries estimate HTLV-1 and HIV-1 co-infection prevalences from 1.55% to 3.9%, highlighting the importance of the HTLV-1 sexual and vertical transmission, but also blood transfusion. For example, a co-infection rate of 3.45% was reported in inmate females from Mozambique [46]; similarly, the prevalence of HIV-1/HTLV-1 was 2.8% among women attending two sexual health clinics in Guinea Bissau [47]. HTLV-1 was also described among HIV-seropositive children (3.9%), suggesting that HTLV-1 is mostly acquired early in life [48]. In contrast, no data on HIV-1/HTLV-1 co-infection was found in the specific situation of Central Africa. However, the only studies available were based either serological survey or seroprevalence evaluation of HTLV-1, HIV-1 and others pathogens in different populations (S1 Table) [49–53].

In summary, we described a high prevalence of HTLV-1 infection among HIV-1 infected patients in rural Gabon. Our study highlights the needs for prevention and care programs for HTLV infection in this region and specific attention to populations that may be co-infected with both retroviruses. Implementing larger investigations and targeting both the general and specific populations will help develop these public health strategies.

## Supporting information

**S1 Fig. Phylogenetic tree generated by the Neighbor-Joining method with a 522bp fragment of the env gene.** Phylogenetic comparisons were performed with the 522-nucleotide *env gp21* gene fragment obtained from 43 HTLV-1 isolates, including the 14 sequences from coinfected HIV-1/HTLV-1 patients (in red) and 29 previously published sequences. The Genbank accession numbers of the new sequences from the coinfected HIV-1/HTLV-1 patients are OL546372- OL546385. The phylogeny was derived by the Neighbor-Joining method with the GTR model. Horizontal branch lengths are drawn to scale, with the bar indicating 0.005 nucleotide replacements per site.
(TIF)

**S2 Fig. HTLV-1 and HIV/HTLV-1 prevalence distribution by sex and age class.** The figure shows the overall prevalence of infection by age group and the prevalence of infection by sex: A) In the HIV-1/HTLV-1 coinfected population; B) In the general population at the same place. Red color indicates global prevalence, yellow and green ones represent prevalence in men and women, respectively.
(TIF)

**S1 Table. Summary of the studies related to HIV-1 and HTLV-1 survey in Central Africa.**
(DOCX)

## Acknowledgments

We thank all patients enrolled in this study, the Ambulatory Treatment Centre (CTA) in Franceville, and the medical unit from the Centre International de Recherches Médicales de Franceville (CIRMF).

## Author Contributions

**Conceptualization:** Augustin Mouinga-Ondémé.

**Data curation:** Ingrid Précilya Koumba Koumba, Antony Idam Mamimandjiami, Abdoulaye Diané, Jéordy Dimitri Engone-Ondo, Landry Erik Mombo.

**Formal analysis:** Larson Boundenga, Jéordy Dimitri Engone-Ondo, Delia Doreen Djuicy.

**Funding acquisition:** Antoine Gessain, Avelin Aghokeng Fobang.

**Investigation:** Larson Boundenga, Ingrid Précilya Koumba Koumba, Antony Idam Mamimandjiami, Abdoulaye Diané.

**Methodology:** Augustin Mouinga-Ondémé.

**Project administration:** Augustin Mouinga-Ondémé.

**Resources:** Jeanne Sica.

**Supervision:** Augustin Mouinga-Ondémé.

**Validation:** Avelin Aghokeng Fobang.

**Writing – original draft:** Augustin Mouinga-Ondémé.

**Writing – review & editing:** Larson Boundenga, Landry Erik Mombo, Antoine Gessain, Avelin Aghokeng Fobang.

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
