## [Decision Letter · Decision Letter 0]

25 Aug 2021

PONE-D-21-19756

Human T-Lymphotropic virus type 1 and Human Immunodeficiency Virus co-infection in rural Gabon

PLOS ONE

Dear Dr. Mouinga-Ondemé,

Thank you for submitting your manuscript to PLOS ONE. After careful consideration, we feel that it has merit but does not fully meet PLOS ONE’s publication criteria as it currently stands. Therefore, we invite you to submit a revised version of the manuscript that addresses the points raised during the review process.

Specifically some technical information (PCR methods, used ELISA, determination of the CD4 count) were missing and deserved to be described in material and method. More importantly, because of the cross-sectional approach a carefull description of the viro-immunological status (point 2 of the reviewer 1) and clinical status of the patient as well as sequence data for the HTLV (point 4 and suggestion of the reviewer 1) deserved to be provided as suggested by the reviewer 1. To note even if not obtained at the time of sampling, clinical data obtained thereafter but for all cases evaluated may provide a tast of longitudinal study.

We look forward to receiving your revised manuscript.

Kind regards,

Pierre Roques, Ph.D.

Academic Editor

PLOS ONE

“NO”

“NO”

6. PLOS requires an ORCID iD for the corresponding author in Editorial Manager on papers submitted after December 6th, 2016. Please ensure that you have an ORCID iD and that it is validated in Editorial Manager. To do this, go to ‘Update my Information’ (in the upper left-hand corner of the main menu), and click on the Fetch/Validate link next to the ORCID field. This will take you to the ORCID site and allow you to create a new iD or authenticate a pre-existing iD in Editorial Manager. Please see the following video for instructions on linking an ORCID iD to your Editorial Manager account: https://www.youtube.com/watch?v=_xcclfuvtxQ.

Additional Editor Comments (if provided):

Please note that only a rewriting will not be sufficient to push up the article to publication. Additional data have to be included as requested by the two reviewers

Reviewers' comments:

Reviewer's Responses to Questions

**Comments to the Author**

1. Is the manuscript technically sound, and do the data support the conclusions?

Reviewer #1: Yes

Reviewer #2: Partly

2. Has the statistical analysis been performed appropriately and rigorously? 

Reviewer #1: Yes

Reviewer #2: I Don't Know

3. Have the authors made all data underlying the findings in their manuscript fully available?

Reviewer #1: Yes

Reviewer #2: No

4. Is the manuscript presented in an intelligible fashion and written in standard English?

Reviewer #1: Yes

Reviewer #2: No

5. Review Comments to the Author

Reviewer #1: In this article by Mouinga-Ondemé et al, the prevalence of HTLV-1 infection was estimated at 7.7% in an HIV-infected population in south-eastern Gabon. Dually infected HIV + HTLV patients have so far been considered at risk of a faster clinical course to AIDS, but data are badly needed to support this hypothesis. Many studies have already been published but Gabon remains one of the right place to carry out such studies due to the superposition of both endemic.

This is of major interest to better understand the factors underlying the progression of HIV in patients infected with HLTV. Unfortunately, the cross sectional aspect of the data collect considerably limits the interest of this paper. The authors proposed that plasma viral load may be slightly lower and CD4 cell count higher in double-infected patients, but these data were not statistically significant.

In addition, before to be considered for publication, various questions remain:

1 / Inclusion criteria for the population infected with HIV in this study. Explain the unbalanced sex ratio. Over representation of the pregnant women ?

2 / EIA + but PCR negative and / or WB not known: additional molecular tests are absolutely necessary in this country with a high STLV / HTLV prevalence. Have you performed additional seological and molecular tests?

3 / The assays used for the RT-PCR viral load and the CD4 count must be specified (manufacturers?)

4 / Could we have more details about HTLV sequencing? Is the population only infected with HTLV-1 B?

5 / Clinical data concerning bi or mono infected patients are essential. HAM / TSP / ATTL reported in dually infected patients? Or more frequent neurological or haematological pathologies in this group?

The study is currently limited by the cross-sectional approach and the scarcity of biological and clinical data.

Suggestion to Authors: Despite the nucleotide diversity between HIV-1 and HTLV-1, their RT enzymes exhibit a fairly similar structure in the palm and fingers. All of these patients are (or have been) treated with 3TC / TFC and 184V and must certainly be present in HIV-RT in the majority of patients. A short sequencing of the HTLV RT pocket could be interesting because the intracellular level of lamivudine will lead to a selection of certain HTLV mutations; V118I and others have been proposed as relevant in explaining the high "resistance" of HTLV to 3TC in the 1980s. But recent interest in resistance to HTLV 3TC at least in vitro (DOI: 10.1086 / 322785) has shown that the question is still open. As the buffy coat was collected, sequencing could be easly performed. The presence of RT 3TC / FTC mutation (s) will confirm a potential replication of HTLV at the cellular level in this doubly infected population and will add original data to this article.

Reviewer #2: 480 / 5000

The article does not have sufficient consistency to support publication in Plos One.

Although the confirmation of HIV/HTLV-1 co-infection in patients from Gabon (Central Africa) is a serious public health problem, from a technical-scientific point of view the objectives are not clearly described, as well as the methodology and, furthermore, the results are insufficient for a publication in Plos One.

Therefore, I recommend that the article be submitted to another journal.

6. PLOS authors have the option to publish the peer review history of their article (what does this mean?). If published, this will include your full peer review and any attached files.

Reviewer #1: No

Reviewer #2: No

---

## [Author Response · Author response to Decision Letter 0]

22 Nov 2021

MOUINGA-ONDÉMÉ Augustin

Unité des Infections Rétrovirales et Pathologies Associées, 

Centre International de Recherches Médicales de Franceville (CIRMF), 

B.P. 769, Franceville, Gabon. ondeme@yahoo.fr

Dear Editor, 

Please find attached a revised version of the manuscript (PONE-D-21-19756) entitled “Human T-Lymphotropic virus type 1 and Human Immunodeficiency Virus co-infection in rural Gabon” by Mouinga-Ondémé and colleagues. We thank you very much for facilitating this set of revisions to our paper.

We appreciate the reviewers’s diligence and we thank them for their constructive and helpful comments on the manuscript. We have modified our manuscript accordingly as detailed below. The main corrections in the new manuscript are highlighted in yellow.

Moreover, the english in the new manuscript was corrected by the native speaker.

We hope that these modifications to the manuscript sufficiently addresses the concerns mentioned by the reviewers and we thank them for their valuable advices.

We hope also that you will now find this manuscript suitable for publication in Plos One. 

Yours Sincerely, 

Augustin Mouinga Ondémé

Letter

Dear Dr. Mouinga-Ondemé,

Thank you for submitting your manuscript to PLOS ONE. After careful consideration, we feel that it has merit but does not fully meet PLOS ONE’s publication criteria as it currently stands. Therefore, we invite you to submit a revised version of the manuscript that addresses the points raised during the review process.

Specifically some technical information (PCR methods, used ELISA, determination of the CD4 count) were missing and deserved to be described in material and method. 

More importantly, because of the cross-sectional approach a carefull description of the viro-immunological status (point 2 of the reviewer 1) and clinical status of the patient as well as sequence data for the HTLV (point 4 and suggestion of the reviewer 1) deserved to be provided as suggested by the reviewer 1. To note even if not obtained at the time of sampling, clinical data obtained thereafter but for all cases evaluated may provide a tast of longitudinal study.

We look forward to receiving your revised manuscript.

Kind regards,

Pierre Roques, Ph.D.

Academic Editor

PLOS ONE

Reviewer 1-

Reviewers' comments:

Reviewer's Responses to Questions

Comments to the Author

1. Is the manuscript technically sound, and do the data support the conclusions?

Reviewer #1: Yes

Reviewer #2: Partly

2. Has the statistical analysis been performed appropriately and rigorously?

Reviewer #1: Yes

Reviewer #2: I Don't Know

3. Have the authors made all data underlying the findings in their manuscript fully available?

Reviewer #1: Yes

Reviewer #2: No

4. Is the manuscript presented in an intelligible fashion and written in standard English?

Reviewer #1: Yes

Reviewer #2: No

Answer to Reviewer's Questions

Reviewer #1: 

In this article by Mouinga-Ondemé et al, the prevalence of HTLV-1 infection was estimated at 7.7% in an HIV-infected population in south-eastern Gabon. Dually infected HIV + HTLV patients have so far been considered at risk of a faster clinical course to AIDS, but data are badly needed to support this hypothesis. Many studies have already been published but Gabon remains one of the right place to carry out such studies due to the superposition of both endemic.

This is of major interest to better understand the factors underlying the progression of HIV in patients infected with HLTV. Unfortunately, the cross sectional aspect of the data collect considerably limits the interest of this paper. The authors proposed that plasma viral load may be slightly lower and CD4 cell count higher in double-infected patients, but these data were not statistically significant.

In addition, before to be considered for publication, various questions remain:

1 / Inclusion criteria for the population infected with HIV in this study. Explain the unbalanced sex ratio. Over representation of the pregnant women ?

In our study, all patients enrolled were HIV-1 infected and receiving antiretroviral therapy (ART) at the only Ambulatory Treatment Centre (ATC) in Haut-Ogooué Province (Franceville is the main city). In other words, our inclusion criterion was receiving treatment at the Centre. We found that the majority of patients followed at the ATC in Franceville were women (209 women and 90 men). This observation is coherent with the study conducted by Liégeois and colleagues in 2012 (http://dx.doi.org/10.7448/IAS.15.2.17985) who showed that the majority of patients followed in the ATC are women. We think this situation may be explained by the fact that men prefer to go to other cities for treatment for fear of being stigmatized. No pregnant women were enrolled in the study. In the discussion section, we raises the concern of vertical transmission of HTLV-1 via breastfeeding in HIV-1/HTLV-1 women, as WHO developed programs to prevent the vertical transmission of HIV from mother to child, especially by using antiretroviral drugs (WHO, 2016).

2 / EIA + but PCR negative and / or WB not known: additional molecular tests are absolutely necessary in this country with a high STLV / HTLV prevalence. Have you performed additional seological and molecular tests?

We thank the reviewer for this question. We used the screening strategy commonly used to detect HTLV-1 and 2 and published in many reports (e.g., Djuicy et al., 2018 https://doi.org/10.1371/journal.pntd.0006832; Ramassamy et al., 2020 doi:10.1111/trf.15838). We acknowledge that other options include additional assays such as the INNO-LIA HTLV I/II Score (Immunogenetics, Gent, Belgium), but we did not do this because we used western blot. In addition, as we discussed in this paper, options are limited for HTLV diagnostic in resource-limited countries. The EIA test used we for serology (HTLV-I/II ELISA 4.0, MP Biomedicals) has high specificity. Because of false EIA+ specimen, we used the WB test for serological confirmation or to discriminate between HTLV-1 and HTLV-2. Without WB results, we consider that PCR results for conserved regions of the genome (tax) and sufficiently different regions of the genome (Env) confirmed or refuted the results of the EIA.

3 / The assays used for the RT-PCR viral load and the CD4 count must be specified (manufacturers?)

As recommended by the reviewer, we added sentences which describe the assays used for the CD4 count and the RT-PCR viral load. Please see Methods section, line 106 to 109 in the new version of manuscript.

4 / Could we have more details about HTLV sequencing? Is the population only infected with HTLV-1 B?

As requested by the reviewers, we provided HTLV sequencing data in the supplementary figure. First, we successfully amplified the PTLV tax region for 23 specimens; however, only 14 specimens had sufficient DNA to amplify a fragment of the Env gene (for more information see the line 135 to 142 of the new version of the manuscript). The phylogenetic tree, in supplementary data, shows that most of the study population is infected with HTLV-1B. Only one patient is infected with HTLV-1D (Gab-2276FCV).

5 / Clinical data concerning bi or mono infected patients are essential. HAM / TSP / ATTL reported in dually infected patients? Or more frequent neurological or haematological pathologies in this group?

We agree with the reviewer that clinical data concerning the study population are essential. None of the participants included in this study presented signs of TSP / HAM and/or ATLL. Unfortunately, we did not have clinical and epidemiological data for both mono and bi infected groups. Only some bacteriological, virological and parasitological opportunistic diseases have been reported in mono and coinfected patients. There were all correlated with the rate of CD4. We did not find it appropriate to discuss this because the purpose of our study was investigated HTLV-1 and HIV-1 co-infection in Haut-Ogooué Province. We agree that this merits further investigations in the future.

The study is currently limited by the cross-sectional approach and the scarcity of biological and clinical data.

Suggestion to Authors: Despite the nucleotide diversity between HIV-1 and HTLV-1, their RT enzymes exhibit a fairly similar structure in the palm and fingers. All of these patients are (or have been) treated with 3TC / TFC and 184V and must certainly be present in HIV-RT in the majority of patients. A short sequencing of the HTLV RT pocket could be interesting because the intracellular level of lamivudine will lead to a selection of certain HTLV mutations; V118I and others have been proposed as relevant in explaining the high "resistance" of HTLV to 3TC in the 1980s. But recent interest in resistance to HTLV 3TC at least in vitro (DOI: 10.1086 / 322785) has shown that the question is still open. As the buffy coat was collected, sequencing could be easly performed. The presence of RT 3TC / FTC mutation (s) will confirm a potential replication of HTLV at the cellular level in this doubly infected population and will add original data to this article.

The suggestion made by the reviewer is very relevant. Unfortunately, plasma and buffy coat collected from these patients, in 2018 and 2019, was also used for several other studies. After sequencing the HTLV Env gene, no more DNA was available from several patients for further investigations. Finally, we were unable to amplify and sequence HTLV RT. We note this approach for our future studies. 

In contrast, we amplified HIV RT to check some mutations in HIV-1/HTLV-1 coinfected patients. The majority of the HIV-1/HTLV-1 co-infected patients had a viral load suppression (VL<2.5 Log); Only one patient had a virological failure (the one co-infected with HTLV-1D). The HIV-1 RT region of 646 bp infected this patient was successfully sequenced. This patient was receiving ABC+3TC+EFV first-line antiretroviral therapy. Several HIV drug resistance mutations (DRMs) were identified: L74I, V75T, Y115F, M184V (NRTI), K103N, Y181C (NNRTI). Please see line 209 to 214.

Reviewer #2: 480 / 5000

The article does not have sufficient consistency to support publication in Plos One.

Although the confirmation of HIV/HTLV-1 co-infection in patients from Gabon (Central Africa) is a serious public health problem, from a technical-scientific point of view the objectives are not clearly described, as well as the methodology and, furthermore, the results are insufficient for a publication in Plos One.

Therefore, I recommend that the article be submitted to another journal.

---

## [Decision Letter · Decision Letter 1]

22 Mar 2022

PONE-D-21-19756R1Human T-Lymphotropic virus type 1 and Human Immunodeficiency Virus co-infection in rural GabonPLOS ONE

Dear Dr. Mouinga-Ondemé,

Thank you for submitting your manuscript to PLOS ONE. After careful consideration, we feel that it has merit but does not fully meet PLOS ONE’s publication criteria as it currently stands. Therefore, we invite you to submit a revised version of the manuscript that addresses the points raised during the review process.

We look forward to receiving your revised manuscript.

Kind regards,

Fatah Kashanchi

Academic Editor

PLOS ONE

Journal Requirements:

Additional Editor Comments:

Please address the Reviewers comments

Reviewers' comments:

Reviewer's Responses to Questions

**Comments to the Author**

1. If the authors have adequately addressed your comments raised in a previous round of review and you feel that this manuscript is now acceptable for publication, you may indicate that here to bypass the “Comments to the Author” section, enter your conflict of interest statement in the “Confidential to Editor” section, and submit your "Accept" recommendation.

Reviewer #1: (No Response)

Reviewer #2: All comments have been addressed

2. Is the manuscript technically sound, and do the data support the conclusions?

Reviewer #1: (No Response)

Reviewer #2: Partly

3. Has the statistical analysis been performed appropriately and rigorously? 

Reviewer #1: (No Response)

Reviewer #2: Yes

4. Have the authors made all data underlying the findings in their manuscript fully available?

Reviewer #1: (No Response)

Reviewer #2: Yes

5. Is the manuscript presented in an intelligible fashion and written in standard English?

Reviewer #1: (No Response)

Reviewer #2: Yes

6. Review Comments to the Author

Reviewer #1: Manuscript PONE-D-21-19756R1, Co-infection with human lymphotropic T virus type 1 and human immunodeficiency virus in rural Gabon is a descriptive cross-sectional study. The lack of clarity regarding the objectives and the lack of basic clinical data without any biological and clinical problems are major limitations to its publication in PlosOne

Despite these major limitations, given the importance of HIV / HTLV co-infection in public health and research and the paucity of studies, it is important to report these epidemiological data to the general public of PlosOne.

We suggest:

- improve the quality of the prevalence of coinfection data by matching the HIV HTLV population with the general population of the same place in the country in order to compare the prevalence of these two retroviruses in these different populations. This could at least lead to showing that the prevalence of HTLV, as expected, is higher in the HIV-infected population.

- and to make this article more attractive, we suggest that the discussion include a review of the literature on this double infection. A summary in the form of a table of epidemiological studies on HIV / HTLV co-infections which comment on the specific situation in Central Africa would be very necessary.

Reviewer #2: In this new version the manuscript presents sufficient consistency to support publication in the Plos One. However some modifications are necessary.

The title of the article sounds appropriate. However the data presented do not support the hypothesis the factors evaluated may favor the rapid evolution and progression to AIDS in HTLV-1/ HIV co-infected patients. This hypothesis must be reviewed since the corresponding author informed to reviewer #1 it is not possible to include new evaluations using the plasma and buffy coat.

Points to improve:

Line 100: Please define ART.

Line 114: Western Blot: please write Western with initial capital letter.

Line 130: Please define ANRS.

METHODS

The methodology should be described more organized and coherent. A good way is to insert subtitles:

1.Study design an population

2.HIV Viral Load and CD4 counts

3.HTLV diagnosis (include serological and molecular diagnosis. e.g.: serological diagnosis was performed..../ Molecular diagnosis was performed for all samples with a positive ELISA result. We extracted viral DNA from buffy-coat. )

4.HIV diagnosis

Line 150: The study was approved ...(It would be better moving this topic to the end of subtitle 1.: “Study design an population”

RESULTS

Patient Population: Please, describe the results in the order they appear in the table 1. This will make it easier for the reader to follow the data.

I suggest adding a bar graph figure of HTLV-1 prevalence PCR among men, women and total, according to the age groups evaluated.

DISCUSSION

It is important to review some points of the discussion after reviewing of the hypothesis that factors evaluated may favor the rapid evolution and progression to AIDS in HTLV-1/ HIV co-infected patients.

7. PLOS authors have the option to publish the peer review history of their article (what does this mean?). If published, this will include your full peer review and any attached files.

Reviewer #1: No

Reviewer #2: No

---

## [Author Response · Author response to Decision Letter 1]

6 May 2022

Dr. MOUINGA-ONDÉMÉ Augustin

Unité des Infections Rétrovirales et Pathologies Associées, 

Centre International de Recherches Médicales de Franceville (CIRMF), 

B.P. 769, Franceville, Gabon. ondeme@yahoo.fr

Dear Editor, 

Please find attached a revised version of the manuscript (PONE-D-21-19756R1) entitled “Human T-Lymphotropic virus type 1 and Human Immunodeficiency Virus co-infection in rural Gabon” by Mouinga-Ondémé and colleagues. We thank you very much for facilitating this set of revisions to our paper.

We appreciate the reviewers’s diligence and we thank them for their constructive and helpful comments on the manuscript. We have modified our manuscript accordingly as detailed below.

Corrections in the new manuscript are highlighted in yellow. Concerning the illustrations, we have added supplementary data: the figure (S2 Fig) and the table (S1 table) as suggested by the reviewers.

We have addressed all the points raised by Reviewers #1 and #2 made the required corrections to the manuscript.

Given the new data analysis, we added a new author with approval from all. Delia Doreen DJUICY is now a co-author of this paper, she was added at 7th place.

We hope that these modifications to the manuscript sufficiently addresses the concerns mentioned by the reviewers and we thank them for their valuable advices.

Yours Sincerely, 

Dr. Augustin Mouinga Ondémé

Letter

PONE-D-21-19756R1

Human T-Lymphotropic virus type 1 and Human Immunodeficiency Virus co-infection in rural Gabon

PLOS ONE

Dear Dr. Mouinga-Ondemé,

Thank you for submitting your manuscript to PLOS ONE. After careful consideration, we feel that it has merit but does not fully meet PLOS ONE’s publication criteria as it currently stands. Therefore, we invite you to submit a revised version of the manuscript that addresses the points raised during the review process.

We look forward to receiving your revised manuscript.

Kind regards,

Fatah Kashanchi

Academic Editor

PLOS ONE

Journal Requirements:

Additional Editor Comments:

Please address the Reviewers comments

Reviewers' comments:

Reviewer's Responses to Questions

Comments to the Author

1. If the authors have adequately addressed your comments raised in a previous round of review and you feel that this manuscript is now acceptable for publication, you may indicate that here to bypass the “Comments to the Author” section, enter your conflict of interest statement in the “Confidential to Editor” section, and submit your "Accept" recommendation.

Reviewer #1: (No Response)

Reviewer #2: All comments have been addressed

2. Is the manuscript technically sound, and do the data support the conclusions?

Reviewer #1: (No Response)

Reviewer #2: Partly

3. Has the statistical analysis been performed appropriately and rigorously?

Reviewer #1: (No Response)

Reviewer #2: Yes

4. Have the authors made all data underlying the findings in their manuscript fully available?

Reviewer #1: (No Response)

Reviewer #2: Yes

5. Is the manuscript presented in an intelligible fashion and written in standard English?

Reviewer #1: (No Response)

Reviewer #2: Yes

6. Review Comments to the Author

Answer to Reviewer's Questions

Reviewer #1

Manuscript PONE-D-21-19756R1, Co-infection with human lymphotropic T virus type 1 and human immunodeficiency virus in rural Gabon is a descriptive cross-sectional study. The lack of clarity regarding the objectives and the lack of basic clinical data without any biological and clinical problems are major limitations to its publication in PlosOne

Despite these major limitations, given the importance of HIV / HTLV co-infection in public health and research and the paucity of studies, it is important to report these epidemiological data to the general public of PlosOne.

We suggest:

Improve the quality of the prevalence of coinfection data by matching the HIV HTLV population with the general population of the same place in the country in order to compare the prevalence of these two retroviruses in these different populations. This could at least lead to showing that the prevalence of HTLV, as expected, is higher in the HIV-infected population.

We agree with the Reviewer for improving the quality of the prevalence. Concerning general population of the same place of the study, prevalence was obtained from data used for our previous study published by Djuicy et al. (doi.org/10.1371/journal.pntd.0006832). A paragraph was added in the discussion section from the lines 281 to 285. Also, a supporting information file, S2 Fig. B, about the HTLV-1 prevalence in the general population, was provided and mentionned at the line 284.

And to make this article more attractive, we suggest that the discussion include a review of the literature on this double infection. A summary in the form of a table of epidemiological studies on HIV / HTLV co-infections which comment on the specific situation in Central Africa would be very necessary.

We thank the Reviewer for this suggestion. At the line 299 to 300, we mentionned that « Although the prevalence of these two retroviruses is very high in many African countries, limited number of studies have been carried on the co-infection.” on the specific situation in Central Africa, we do not find published studies about HIV-1/HTLV-1 co-infection. We found only studies based on serological survey or seroprevalence evaluation of HTLV-1, HIV-1 and others pathogens in different populations; and some of these studies are summarized in the S1 table. Lines 307 to 311.

Reviewer #2

In this new version the manuscript presents sufficient consistency to support publication in the Plos One. However some modifications are necessary.

The title of the article sounds appropriate. However the data presented do not support the hypothesis the factors evaluated may favor the rapid evolution and progression to AIDS in HTLV-1/ HIV co-infected patients. This hypothesis must be reviewed since the corresponding author informed to reviewer #1 it is not possible to include new evaluations using the plasma and buffy coat.

Points to improve:

- Line 100: Please define ART.

As recommended by the Reviewer, ART was defined as « antiretroviral therapy » (line 101 in the new manuscript version).

- Line 114: Western Blot: please write Western with initial capital letter.

As recommended by the Reviewer, « Western » was written with initial capital letter (line 132).

- Line 130: Please define ANRS.

We agree with the Reviewer, and ANRS was defined as « French National Agency for Research on AIDS and Viral Hepatitis. » (lines 122 to 123).

METHODS

The methodology should be described more organized and coherent. A good way is to insert subtitles:

1.Study design an population 

2.HIV Viral Load and CD4 counts 

3.HTLV diagnosis (include serological and molecular diagnosis. e.g.: serological diagnosis was performed..../ Molecular diagnosis was performed for all samples with a positive ELISA result. We extracted viral DNA from buffy-coat. ) 

4.HIV diagnosis

We agree with the Reviewer. The METHODS section was edited and organized as recommended :

1. Study design and population (Line 96) 

2. HIV Viral Load and CD4 counts (Line 107)

3. HIV diagnosis (Line 116)

4. HTLV diagnosis (Line 127)

5. Phylogenetic analysis (Line154)

6. Statistical analysis (Line 160)

- Line 150: The study was approved ...(It would be better moving this topic to the end of subtitle 1.: “Study design an population”

As recommended by the Reviewer, this topic was moved to the end of « Study design and population » section (lines 104 to 105)

RESULTS

- Patient Population: Please, describe the results in the order they appear in the table 1. This will make it easier for the reader to follow the data. 

We agree with the Reviewer. Results was organized by following the order they appear in the table 1 (lines 173 to 175)

I suggest adding a bar graph figure of HTLV-1 prevalence PCR among men, women and total, according to the age groups evaluated. 

As suggested by the Reviewer, a bar graph has been added as a supplementary data (S2 Fig. A), and this reference was inserted in the main text (line 218).

DISCUSSION

It is important to review some points of the discussion after reviewing of the hypothesis that factors evaluated may favor the rapid evolution and progression to AIDS in HTLV-1/ HIV co-infected patients.

We agree with the Reviewer. In the discussion section, a paragraph was added at the lines 270 to 277

7. PLOS authors have the option to publish the peer review history of their article (what does this mean?). If published, this will include your full peer review and any attached files.

Do you want your identity to be public for this peer review? For information about this choice, including consent withdrawal, please see our Privacy Policy.

Reviewer #1: No

Reviewer #2: No

---

## [Decision Letter · Decision Letter 2]

29 Jun 2022

Human T-Lymphotropic virus type 1 and Human Immunodeficiency Virus co-infection in rural Gabon

PONE-D-21-19756R2

Dear Dr. Mouinga-Ondemé,

We’re pleased to inform you that your manuscript has been judged scientifically suitable for publication and will be formally accepted for publication once it meets all outstanding technical requirements.

Kind regards,

Fatah Kashanchi

Academic Editor

PLOS ONE

Additional Editor Comments (optional):

Reviewers' comments:

Reviewer's Responses to Questions

**Comments to the Author**

1. If the authors have adequately addressed your comments raised in a previous round of review and you feel that this manuscript is now acceptable for publication, you may indicate that here to bypass the “Comments to the Author” section, enter your conflict of interest statement in the “Confidential to Editor” section, and submit your "Accept" recommendation.

Reviewer #1: (No Response)

2. Is the manuscript technically sound, and do the data support the conclusions?

Reviewer #1: Partly

3. Has the statistical analysis been performed appropriately and rigorously? 

Reviewer #1: Yes

4. Have the authors made all data underlying the findings in their manuscript fully available?

Reviewer #1: (No Response)

5. Is the manuscript presented in an intelligible fashion and written in standard English?

Reviewer #1: Yes

6. Review Comments to the Author

Reviewer #1: (No Response)

7. PLOS authors have the option to publish the peer review history of their article (what does this mean?). If published, this will include your full peer review and any attached files.

Reviewer #1: No

---

## [Editor Report · Acceptance letter]

13 Jul 2022

PONE-D-21-19756R2 

Human T-Lymphotropic virus type 1 and Human Immunodeficiency Virus co-infection in rural Gabon 

Dear Dr. Mouinga-Ondémé:

I'm pleased to inform you that your manuscript has been deemed suitable for publication in PLOS ONE. Congratulations! Your manuscript is now with our production department. 

Kind regards, 

on behalf of

Dr. Fatah Kashanchi 

Academic Editor

PLOS ONE